# Towards Reliable Marking and Verification of AI-Generated Text via Geometry-aware Sentence-level Watermarking

**Yubing Ren** [1 2]  **Ping Guo** [3]  **Yanan Cao** [1 2]

## Abstract

Large generative models raise growing concerns about provenance, misinformation, and impersonation. Digital watermarking offers a principled solution, yet extending it to natural language remains challenging due to text discreteness and sensitivity to semantic perturbations. Existing text watermarking methods either operate at the token level requiring white-box access and remaining fragile to paraphrasing, or at the sentence level, which supports black-box deployment but suffers from low Watermark Success Rate (WSR). We show that low WSR in sentence-level watermarking primarily stems from low injection success probability caused by a mismatch between posterior embedding distributions and semantic accept regions. Based on this insight, we propose **X-Guard**, a geometry-aware sentence-level watermarking framework that improves injection success by systematically optimizing embedding distributions and semantic space partitioning. X-Guard learns a more isotropic embedding space and introduces $\mathbf{A^2PQ}$, a centroid-aligned partitioning scheme that approximately equalizes probability mass across regions. Extensive experiments across multiple models, languages, and attack settings demonstrate that X-Guard consistently improves robustness while preserving text fluency and practical deployability. Code and data are available at https://github.com/lilice-r/X-Guard.

[1]Institute of Information Engineering, Chinese Academy of Sciences, Beijing, China [2]School of Cyber Security, University of Chinese Academy of Sciences, Beijing, China [3]Independent Reseacher. Correspondence to: Yanan Cao <caoyanan@iie.ac.cn>.

## 1. Introduction

Recent advances in generative models, such as GPT-5 (OpenAI, 2025), DALL·E 3 (OpenAI, 2023), Stable Diffusion 3.5 (AI, 2024), and VALL-E 2 (Chen et al., 2024b), have enabled highly realistic content generation across text, image, audio, and video modalities. While these models power real-world applications, they also introduce new risks, including misinformation, impersonation, and content forgery. Ensuring reliable provenance and accountability for AI-generated content has therefore become a critical challenge.

Digital watermarking offers a promising mechanism for distinguishing between AI-generated and human-authored content. A watermarking system consists of two stages: *injection*, which embeds hidden signals during generation, and *verification*, which determines whether such signals are present. Although watermarking has been widely studied for continuous media, extending it to natural language remains challenging due to text discreteness, low redundancy, and sensitivity to small semantic or lexical perturbations.

An effective text watermarking scheme should satisfy three key requirements: *reliability*, *imperceptibility*, and *universality*. The watermark should remain detectable under common transformations such as paraphrasing, translation, or adversarial edits, while preserving text fluency and semantic fidelity. Moreover, practical deployment requires compatibility with both white-box and black-box models, as well as robustness across multiple languages.

Existing text watermarking approaches can be broadly categorized into *token-level* and *sentence-level* methods. Token-level watermarking embeds signals by biasing token sampling or applying deterministic lexical edits. While probabilistic methods can achieve high detection accuracy, they rely on access to internal model states and are unsuitable for black-box settings. Rule-based token edits, on the other hand, are fragile and can be easily disrupted by surface-level modifications such as paraphrasing or synonym substitution.

To address these limitations, recent work has shifted toward *sentence-level watermarking*, which embeds watermarks in a continuous semantic feature space rather than directly manipulating surface tokens. By defining watermarkable regions in the embedding space and rejecting sentences

that fall outside these regions, sentence-level methods naturally support black-box deployment and exhibit stronger robustness to meaning-preserving transformations. However, existing sentence-level approaches often suffer from low and unstable *Watermark Success Rate (WSR)*, especially under multilingual or adaptive attacks.

**Key Insight.** We observe that the effectiveness of sentence-level watermarking is fundamentally governed by the geometry of the underlying semantic space. In practice, sentence encoders induce anisotropic and uneven embedding distributions, causing most probability mass to concentrate in a small subset of regions. As a result, randomly selected watermark regions often poorly align with the model's posterior embedding distribution, leading to low *injection success probability*. When the model's natural semantic outputs fall outside the watermark region, watermark injection fails regardless of detection quality, ultimately limiting WSR.

**Our Work.** We present **X-Guard**, a sentence-level watermarking framework that explicitly addresses this geometric mismatch by systematically optimizing the embedding distribution and the semantic space partitioning. We show that the primary bottleneck of existing methods lies in low injection success probability, rather than insufficient detection sensitivity. To mitigate this issue, X-Guard improves watermark injection from two complementary perspectives. First, we introduce an embedding training strategy that promotes a more spherical and isotropic posterior embedding distribution, expanding the effective semantic space available for watermarking. Second, we propose **$A^2$PQ**, an embedding-aware partitioning mechanism whose hyperplane boundaries are aligned with the centroid of the posterior distribution and approximately equalize probability mass across cells. Together, these designs nearly double the injection success probability compared to prior sentence-level methods.

For verification, X-Guard analyzes the statistical distribution of sentence embeddings to determine watermark presence. We evaluate X-Guard across multiple languages, model architectures, and threat settings, including both white-box and black-box deployment. Results demonstrate that X-Guard achieves reliable, imperceptible, and robust watermarking across diverse linguistic and practical scenarios.

## 2. Background and Related Works

### 2.1. Token-level Text Watermarking

● *Rule-based Methods,* embed watermarks through surface edits, such as Unicode encoding or homoglyph manipulation (Por et al., 2012; Rizzo et al., 2016; Sato et al., 2023), synonym substitutions (Yang et al., 2022; Topkara et al., 2006b; Yang et al., 2023; Yoo et al., 2023), or syntactic rewrites (Atallah et al., 2001; Topkara et al., 2006a). Handcrafted rules are fragile and can be easily disrupted by para-

phrasing, translation, or reformatting.

● *Neural-based Methods,* jointly train language models to encode hidden messages in their generated token sequences (Abdelnabi & Fritz, 2021; Zhang et al., 2024; Lau et al., 2024). Although these methods operate at a higher level of abstraction and embed watermarks implicitly in model parameters, the resulting signals still manifest through token-level lexical or syntactic patterns, rather than semantic or sentence-level representations.

● *Inference-time Methods,* inject watermark by biasing the token generation. KGW (Kirchenbauer et al., 2023) partitions the vocabulary into "green" and "red" sets to bias token selection, later refined through context- or semantic-aware hashing (Kirchenbauer et al., 2024; Zhao et al., 2024; Ren et al., 2024a; He et al., 2024; Liu et al., 2024; Liu & Bu, 2024; Lee et al., 2024; Lu et al., 2024; Ren et al., 2024b; Wang et al., 2025; 2026a;b; Li et al., 2026). Others improve stealthiness through randomized or contrastive decoding (Aaronson, 2023; Christ et al., 2024; Fu et al., 2024; Kuditipudi et al., 2024; Zhu et al., 2024; Dathathri et al., 2024).

### 2.2. Sentence-level Text Watermarking

The dominant paradigm is *negative-sampling* methods, which first encode each sentence, partition the semantic space into discrete cells, and designate a subset of cells as *accepted region*. Watermark embedding is then achieved by repeatedly sampling sentences until their embeddings fall into accepted cells. SemStamp (Hou et al., 2024a) is most representative, but its partitioning strategy is based on random LSH, disregarding the geometry and anisotropy of the embedding distribution. The follow-up k-SemStamp (Hou et al., 2024b) replaces LSH with $k$-means clustering, yet it still relies on *prior* embedding distributions. Recent methods (Dabiriaghdam & Wang, 2025; Huo et al., 2025) attempt to incorporate posterior embedding distribution when constructing partitions, but their accept region selection is cherry picking, sensitive to the choice of encoder.

### 2.3. Product Quantization

Product Quantization (PQ) (Jégou et al., 2011) is a method for partitioning a high-dimensional feature space into compact regions. It learns data-dependent boundaries based on the distribution of existing embeddings.

Given a set of embedding vectors $\{\mathbf{z}_i \in \mathbb{R}^d\}_{i=1}^S$, PQ first splits each vector into $M$ equal-length subvectors:

$$\mathbf{z}_i = [\mathbf{z}_i^{(1)}, \dots, \mathbf{z}_i^{(M)}], \quad \mathbf{z}_i^{(m)} \in \mathbb{R}^{d/M}. \quad (1)$$

For each subspace $m$, PQ performs $K$-means clustering on all $\{\mathbf{z}_i^{(m)}\}_{i=1}^S$ to obtain $K$ centroids: $\mathcal{C}^{(m)} =$

$$\{\mathbf{c}_1^{(m)}, \ldots, \mathbf{c}_K^{(m)}\}.$$

These centroids form the codebook of the $m$-th subspace and divide it into $K$ local regions that reflect the empirical embedding distribution. Combining all $M$ subspaces yields a global partition of the space into $K^M$ distinct cells.

Given a new embedding $\mathbf{z}'$, PQ assigns each subvector to its nearest centroid:

$$\mathrm{idx}^{(m)}(\mathbf{z}'^{(m)}) = \arg\min_{j \in [K]} \|\mathbf{z}'^{(m)} - \mathbf{c}_j^{(m)}\|_2. \quad (2)$$

The indices $\mathrm{idx}^{(1)}, \ldots, \mathrm{idx}^{(M)}$ together specify which cell $\mathbf{z}'$ belongs to:

$$\mathrm{PQ}(\mathbf{z}') = [\mathrm{idx}^{(1)}, \ldots, \mathrm{idx}^{(M)}]. \quad (3)$$

## 3. Methodology

### 3.1. Sentence-level Watermarking Scheme

Sentence-level watermarking embeds signals in a semantic feature space that is stable under paraphrasing and lexical variation, providing robustness to meaning-preserving edits and compatibility with black-box deployment. Given an input $\mathbf{x}$, the model generates a sentence $\mathbf{y}_t$ at step $t$, which is encoded as $\mathbf{z}_t = E(\mathbf{y}_t) \in \mathbb{R}^d$. The semantic space $\Omega$ is partitioned into disjoint cells $\{C_1, \ldots, C_{|\Omega|}\}$. A secret key selects an accept region $\mathcal{R}_{\mathrm{acc}} \subset \Omega$ with $|\mathcal{R}_{\mathrm{acc}}| = \gamma|\Omega|$.

A sentence is accepted if $\mathbf{z}_t \in \mathcal{R}_{\mathrm{acc}}$, otherwise it is rejected and resampled up to $T$ attempts, enforcing generation within a watermarkable subspace. Robustness arises from embedding stability: meaning-preserving edits yield nearby embeddings that remain within the same accept region, which is encouraged via contrastive training of the encoder.

### 3.2. Injection Success Probability

Existing sentence-level methods often exhibit unstable WSR across prompts and languages. Beyond verification sensitivity, we identify a more fundamental bottleneck: *injection* can fail intrinsically when the model's natural generations rarely fall into the accept region. We therefore analyze the *Injection Success Probability* $p_{\mathrm{is}}$ by modeling the probability that a naturally generated embedding lies in $\mathcal{R}_{\mathrm{acc}}$. Sentence-level schemes partition the semantic space $\Omega$ into disjoint cells $\{C_1, \ldots, C_{|\Omega|}\}$. Let $p(\mathbf{z})$ be the (intractable) density of sentence embeddings $\mathbf{z} = E(\mathbf{y})$. We define the probability mass of each cell:

$$\mu_j = \int_{C_j} p(\mathbf{z})\, d\mathbf{z}, \quad \text{with} \quad \sum_{j \in \Omega} \mu_j = 1. \quad (4)$$

In practice, we estimate $\mu_j$ via sampling: for a fixed prompt, we draw $N$ stochastic continuations $\{\mathbf{y}_i\}_{i=1}^N$ (nucleus sam-

pling), encode $\mathbf{z}_i = E(\mathbf{y}_i)$, and compute

$$\hat{\mu}_j = \frac{1}{N} \sum_{i=1}^N \mathbf{1}[\mathbf{z}_i \in C_j]. \quad (5)$$

Unless otherwise specified, we use $N = 400$.

**Acceptance probability.** Given an accept region $\mathcal{R}_{\mathrm{acc}}$ (a key-selected subset of cells with $|\mathcal{R}_{\mathrm{acc}}| = n_{\mathrm{acc}} = \gamma|\Omega|$), the probability that a *single* natural generation is accepted equals the total mass of selected cells:

$$p_{\mathrm{nat}}(\mathcal{R}_{\mathrm{acc}}) = \sum_{C_j \in \mathcal{R}_{\mathrm{acc}}} \mu_j. \quad (6)$$

**Injection success with resampling.** With up to $T$ independent resampling attempts, injection succeeds if at least one sample falls in $\mathcal{R}_{\mathrm{acc}}$. Thus,

$$p_{\mathrm{is}}(T \mid \mathcal{R}_{\mathrm{acc}}) = 1 - (1 - p_{\mathrm{nat}}(\mathcal{R}_{\mathrm{acc}}))^T. \quad (7)$$

Finally, since $\mathcal{R}_{\mathrm{acc}}$ is formed by uniformly selecting $n_{\mathrm{acc}}$ cells from $\Omega$, the expected injection success probability is

$$p_{\mathrm{is}}(T) = \frac{1}{\binom{|\Omega|}{n_{\mathrm{acc}}}} \sum_{\mathcal{R}_{\mathrm{acc}} \subseteq \Omega} \left[ 1 - (1 - p_{\mathrm{nat}}(\mathcal{R}_{\mathrm{acc}}))^T \right]. \quad (8)$$

**Implication: mass imbalance yields low $p_{\mathrm{is}}$.** When the embedding distribution is highly imbalanced across cells (i.e., a few cells dominate $\mu_j$), a randomly selected $\mathcal{R}_{\mathrm{acc}}$ is unlikely to capture sufficient probability mass, making $p_{\mathrm{nat}}(\mathcal{R}_{\mathrm{acc}})$ small and $p_{\mathrm{is}}$ low even for moderate $T$. For a toy example with $|\Omega| = 4$, $\gamma = 0.25$ (thus $n_{\mathrm{acc}} = 1$), and estimated masses $(\mu_1, \mu_2, \mu_3, \mu_4) = (0, 0.04, 0.9, 0.06)$, we obtain $p_{\mathrm{is}}(T{=}10) = 44.9\%$, showing that substantial resampling can still fail to watermark roughly half of natural generations under severe mass imbalance.

### 3.3. Key Ideas of Our Design

Motivated by the analysis in Sec. 3.2, our design aims to improve the uniformity of cell-wise probability mass $\mu_j$ from two complementary perspectives: (i) reshaping the embedding distribution to be more isotropic, and (ii) constructing centroid-aligned, distribution-aware semantic partitions. Together, these designs increase injection success probability and stabilize sentence-level watermarking.

First, we optimize the encoder to produce embeddings that are both semantically aligned and globally uniform. As illustrated in Fig. 1b, the resulting embedding distribution is substantially more spherical than prior methods (Fig. 1a), yielding a more even allocation of probability mass across the semantic space. Second, we introduce an embedding-aware partitioning scheme, termed *Anchor Angular Product Quantization* ($A^2$PQ), whose hyperplane boundaries follow the centroid of the posterior embedding distribution, leading to more balanced $\mu_j$ values.

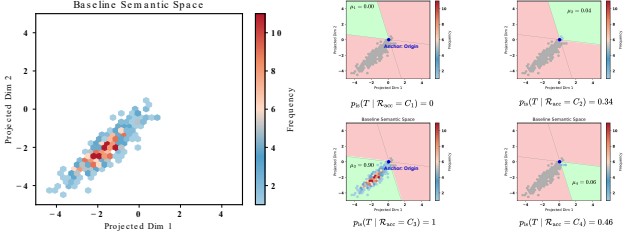

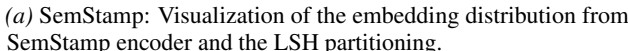

*(a)* SemStamp: Visualization of the embedding distribution from SemStamp encoder and the LSH partitioning.

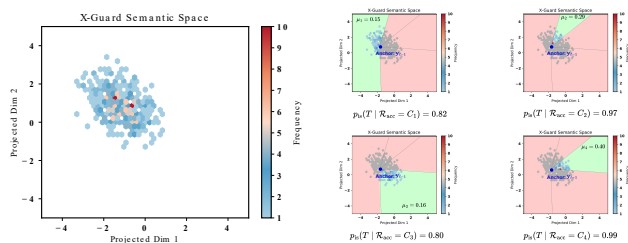

*(b)* X-Guard: Visualization of the embedding distribution from X-Guard encoder and our $A^2PQ$ partitioning.

*Figure 1.* t-SNE visualization of semantic space. For each subfigure, the left panel visualizes the posterior distribution of all naturally generated sentences $\mathbf{y}$, conditioned on a fixed prompt $\mathbf{x}$, within the encoder's semantic space $\Omega$. Fig. 1b highlights: (1) a more spherical posterior distribution that exhibits improved isotropy, and (2) a distribution-aware partitioning scheme whose hyperplane boundaries follow the centroid of the posterior distribution and equalize the probability mass $\mu$ across cells.

### 3.3.1. OPTIMIZING ENCODER

Effective sentence-level watermarking requires an encoder that preserves semantic similarity while avoiding highly anisotropic embedding distributions. We achieve this by training the encoder with the Alignment–Uniformity (A&U) objective (Wang & Isola, 2020), which explicitly decouples semantic alignment from distributional uniformity. The training loss is defined as

$$\mathcal{L}_{A\&U} = \mathcal{L}_{align} + \lambda \mathcal{L}_{uniform}. \tag{9}$$

The alignment term enforces semantically equivalent sentences to have nearby embeddings,

$$\mathcal{L}_{align} = \mathbb{E}_{(\mathbf{y}, \mathbf{y}^+)} \|\mathbf{z} - \mathbf{z}^+\|_2^2, \tag{10}$$

where $\mathbf{z}$ and $\mathbf{z}^+$ are L2-normalized embeddings of a positive pair. The uniformity term encourages embeddings to spread evenly over the unit hypersphere,

$$\mathcal{L}_{uniform} = \log \mathbb{E}_{(\mathbf{y}_i, \mathbf{y}_j) \sim P_{data}} \left[ e^{-2 \|\mathbf{z}_i - \mathbf{z}_j\|_2^2} \right]. \tag{11}$$

Together, these objectives produce embeddings that are both semantically stable under paraphrasing and more evenly distributed, directly improving the balance of $\mu_j$.

### 3.3.2. ANCHOR ANGULAR PRODUCT QUANTIZATION

While encoder optimization improves global uniformity, partitioning must further align with the posterior embedding geometry. Conventional methods often induce severe mass imbalance, with most embeddings concentrated in a few cells. To address this, $A^2PQ$ is designed with two principles: (i) partition hyperplanes intersect at the centroid of the posterior distribution, and (ii) probability mass is approximately equalized across cells.

Because the exact centroid of next-step embeddings is intractable, we use the previous-step embedding $\mathbf{z}^{(t-1)}$ as a local semantic anchor. Given the encoder's Lipschitz continuity, $\mathbf{z}^{(t-1)}$ provides a reliable proxy for the centroid. We define the anchor-relative residual as $\mathbf{r}_t = \mathbf{z}_t - \mathbf{z}_{t-1}$.

To construct partitions that share a common intersection at the anchor while remaining distribution-aware, $A^2PQ$ adopts Product Quantization with an angular (cosine) metric. Unlike standard PQ, which clusters subvectors using Euclidean distance, $A^2PQ$ performs $k$-means under cosine distance, clustering by direction rather than magnitude. For the $m$-th subvector, assignment is given by

$$\text{idx}^{(m)}(\mathbf{r}^{(t,m)}) = \arg\max_{j \in [K]} \frac{\langle \mathbf{r}_t^{(m)}, \mathbf{c}_j^{(m)} \rangle}{\|\mathbf{r}_t^{(m)}\|_2 \|\mathbf{c}_j^{(m)}\|_2}. \tag{12}$$

As shown in Fig. 1b, $A^2PQ$ yields more balanced partitions. Under the improved embedding distribution $(\mu_0, \mu_1, \mu_2, \mu_3) = (0.15, 0.29, 0.16, 0.40)$ with $|\Omega| = 4$, $n_{acc} = 1$, and $T = 10$, the Injection Success Probability increases to 89.9%, compared to 44.9% under the baseline.

## 3.4. Design Details of X-Guard

### 3.4.1. EMBEDDING DISTRIBUTION FINE-TUNING

We fine-tune an encoder $E(\cdot)$ to produce embeddings that align semantically equivalent sentences while yielding a balanced embedding distribution tailored to LLM outputs and their adversarial variants. We curate a high-quality sentence pool from CC-Net (Wenzek et al., 2020), label each sentence with a domain tag using NVIDIA's domain classifier, and uniformly sample 200 sentences per domain, resulting in 5,200 English seed sentences. Each sentence is truncated to 50 tokens and used as a prompt to the LLM, whose continuation is extracted for training. We construct two types of positive pairs. For *monolingual pairs*, we paraphrase each sentence using `gpt-3.5-turbo`. For *multilingual pairs*, we translate each sentence into over 50 languages to form parallel data. The encoder is trained using the Alignment–Uniformity (A&U) contrastive loss. Hyper-parameters are reported in Sec. 4.1. Sentence embeddings are computed as the mean of final-layer token representations and L2-normalized for subsequent injection.

### 3.4.2. SEMANTIC SPACE PARTITIONING

We construct the semantic partition using $A^2PQ$ on embeddings produced by the encoder. Using the same domain-labeled prompts, we generate ten diverse continuations per prompt via temperature decoding. Each continuation is segmented into sentences and encoded as $\mathbf{z}$. For each adjacent sentence pair, we compute the anchor-relative residual $\mathbf{r}_t = \mathbf{z}_t - \mathbf{z}_{t-1}$, yielding over 100,000 residual vectors that characterize semantic transitions.

Each residual $\mathbf{r}_t \in \mathbb{R}^d$ is divided into $M$ equal-length subvectors $\mathbf{r}_t = [\mathbf{r}_t^{(1)}, \ldots, \mathbf{r}_t^{(M)}]$, with $\mathbf{r}_t^{(m)} \in \mathbb{R}^{d/M}$. For each subspace $m$, we cluster $\{\mathbf{r}_t^{(m)}\}$ using $K$-means under cosine distance. The resulting centroids $\mathcal{C}^{(m)} = \{\mathbf{c}_1^{(m)}, \ldots, \mathbf{c}_K^{(m)}\}$ form the codebook for subspace $m$.

The Cartesian product of all subspace indices defines the partition $\Omega = \{C_1, \ldots, C_{|\Omega|}\}$ with $|\Omega| = K^M$, where each cell $C_j = (\text{idx}^{(1)}, \ldots, \text{idx}^{(M)})$ corresponds to a semantic region with empirical probability mass $\mu_j$. All codebooks and mappings are stored as the partition signature and used for both injection and verification.

### 3.4.3. DESIGN DETAILS OF INJECTION FUNCTION

At generation step $t$, given the prefix $\mathbf{y}_{t-1}$, we compute its residual embedding $\mathbf{r}_{t-1} = E(\mathbf{y}_{t-1}) - E(\mathbf{y}_{t-2})$. Each subvector $\mathbf{r}_t^{(m)}$ is assigned to its nearest angular codeword:

$$\text{idx}^{(m)}(\mathbf{r}_{t-1}^{(m)}) = \arg\max_{j \in [K]} \frac{\langle \mathbf{r}_{t-1}^{(m)}, \mathbf{c}_j^{(m)} \rangle}{\|\mathbf{r}_{t-1}^{(m)}\|_2 \|\mathbf{c}_j^{(m)}\|_2}. \tag{13}$$

The resulting signature $\text{sig}_{t-1} = (\text{idx}^{(1)}, \ldots, \text{idx}^{(M)})$, and a secret key $k$ are derived for a deterministic pseudo-random seed: $\text{seed}_t = \text{Hash}(k \,\|\, \text{sig}_{t-1})$, where $\text{Hash}(\cdot)$ is a cryptographic hash or keyed PRF. The seed defines a deterministic permutation of all cells in $\Omega$, and the accept region $\mathcal{R}_{\text{acc}}^t$ is formed by selecting the first $n_{\text{acc}} = \gamma|\Omega|$ cells. Unlike token-level schemes whose seeds depend on raw tokens and are fragile to edits, our seed is derived from the embedding $\text{sig}_{t-1}$, making the accept region recoverable under meaning-preserving attacks.

Given a candidate sentence $\mathbf{y}_t$, we compute $\mathbf{z}_t = E(\mathbf{y}_t)$ and the residual $\mathbf{r}_t = \mathbf{z}_t - \mathbf{z}_{t-1}$. And using the same method in Equation 13 to caluclate signature $\text{sig}_t$, which determines acceptance: if $\text{sig}_t \in \mathcal{R}_{\text{acc}}^t$, the sentence is accepted; otherwise, it is rejected and resampled until acceptance or a maximum of $T$ attempts.

This injection process is deterministic given $(k, \mathbf{z}_{t-1})$ and the codebooks, ensuring stable watermark embedding while remaining robust to adversarial perturbations of the prefix.

---

**Algorithm 1** X-Guard Watermark Inject (step $t$)

**Input:** LM $p_\theta$, encoder $E$, codebooks $\{\mathcal{C}^{(m)}\}_{m=1}^M$, secret key $k$, prefix residual $\mathbf{r}_{t-1}$; ratio $\gamma \in (0, 1)$; max trials $T$.
**Output:** Continuation $\mathbf{y}_t$.
1: **for** $m = 1$ to $M$ **do**
2:      $\text{idx}^{(m)} \leftarrow \arg\max_{j \in [K]} \dfrac{\langle \mathbf{r}_{t-1}^{(m)}, \mathbf{c}_j^{(m)} \rangle}{\|\mathbf{r}_{t-1}^{(m)}\|_2 \|\mathbf{c}_j^{(m)}\|_2}$
3: **end for**
4: $\text{sig}_{t-1} \leftarrow (\text{idx}^{(1)}, \ldots, \text{idx}^{(M)})$
5: $\text{seed}_t \leftarrow \text{Hash}(k\|\mathbf{z}_{t-1})$
6: $n_{\text{acc}} \leftarrow \lfloor \gamma \cdot |\Omega| \rfloor$
7: $\pi \leftarrow \text{PRNG\_perm}(|\Omega|; \text{seed}_t)$
8: $\mathcal{R}_{\text{acc}}^t \leftarrow \{\pi(1), \ldots, \pi(n_{\text{acc}})\}$
9: **for** $\ell = 1$ to $T$ **do**
10:      $\mathbf{y}_t \sim p_\theta(\cdot \mid \mathbf{y}_{t-1})$
11:      $\mathbf{z}_t \leftarrow E(\mathbf{y}_t)$
12:      $\mathbf{r}_t \leftarrow \mathbf{z}_t - \mathbf{z}_{t-1}$
13:      **for** $m = 1$ to $M$ **do**
14:          $\text{idx}^{(m)} \leftarrow \arg\max_{j \in [K]} \dfrac{\langle \mathbf{r}_t^{(m)}, \mathbf{c}_j^{(m)} \rangle}{\|\mathbf{r}_t^{(m)}\|_2 \|\mathbf{c}_j^{(m)}\|_2}$
15:      **end for**
16:      $\text{sig}_t \leftarrow (\text{idx}^{(1)}, \ldots, \text{idx}^{(M)})$
17:      **if** $\text{sig}_t \in \mathcal{R}_{\text{acc}}^t$ **then**
18:          **return** $\mathbf{y}_t$
19:      **end if**
20:      $last \leftarrow \mathbf{y}_t$
21: **end for**
22: **return** $last$

---

### 3.4.4. WATERMARK VERIFICATION

X-Guard adopts a $z$-test-based hypothesis testing framework, which requires aggregating evidence across multiple sentences to control false positives. Given a text consisting of $S$ sentences $\{\mathbf{y}_1, \ldots, \mathbf{y}_S\}$, each sentence is encoded as $\mathbf{z}_i = E(\mathbf{y}_i)$. Using the same secret key and anchor-based seed derivation as in injection, we reconstruct the corresponding accept region $\mathcal{R}_{\text{acc}}^i$ for each sentence. We then compute the quantized signature $\text{sig}(\mathbf{y}_i)$ and count the number of sentences whose signatures fall into their accept regions, denoted as $S_W$.

Under the null hypothesis $H_0$ (unwatermarked text), each sentence independently falls into an accept region with probability $\gamma$. We compute the $z$-score:

$$z_{\text{X-Guard}} = \frac{S_W - \gamma \cdot S}{\sqrt{\gamma \cdot (1 - \gamma) \cdot S}}, \tag{14}$$

and classify the text as *watermarked* if $z_{\text{X-Guard}} > Z_r$. The minimum number of sentences required for verification is:

$$S_{\min} = \frac{Z_r^2 \, \gamma(1 - \gamma)}{\left(\dfrac{S_W}{S} - \gamma\right)^2}, \tag{15}$$

where $S_W/S$ is the empirical detection success rate. We set $Z_r = 4$ and $\gamma = 0.25$. Without attack, $S_W/S$ can be

*Table 1.* Comparison of watermark robustness across low-res and high-res languages under different attack settings on LLaMA3.2-3B.

| Method | No Attack | | Substitution 10% | | Substitution 30% | | Mono-ling Paraphrase | | Cross-ling Paraphrase | | Translation | | Optimized Adaptive | |
|---|---|---|---|---|---|---|---|---|---|---|---|---|---|---|
| | WSR | AUC | WSR | AUC | WSR | AUC | WSR | AUC | WSR | AUC | WSR | AUC | WSR | AUC |
| **Low-resource Languages (ar, ms, th, hu)** | | | | | | | | | | | | | | |
| *Token-level Watermark Method* | | | | | | | | | | | | | | |
| KGW | 91.85% | 99.04% | 74.23% | 77.53% | 56.00% | 59.30% | 0.55% | 51.90% | 0.28% | 47.90% | 0.28% | 47.85% | 0.15% | 45.31% |
| Uni-WM | 88.40% | 98.86% | 81.53% | 83.33% | 69.05% | 70.85% | 1.95% | **75.18%** | 0.75% | 59.65% | 0.75% | 55.50% | 0.35% | 52.33% |
| SynthID | 94.80% | **99.59%** | 84.60% | 84.90% | 75.53% | 75.83% | 1.20% | 49.88% | 1.60% | 49.94% | 1.55% | 49.94% | 0.45% | 40.47% |
| *Sentence-level Watermark Method* | | | | | | | | | | | | | | |
| SemStamp | 64.80% | 89.58% | 62.10% | 86.90% | 60.79% | 79.57% | 26.69% | 64.88% | 31.95% | 61.65% | 32.83% | 61.98% | 25.30% | 59.10% |
| SimMark | 66.93% | 77.60% | 65.21% | 78.56% | 65.88% | 75.43% | 27.94% | 63.41% | 30.88% | 62.97% | 34.62% | 60.71% | 26.85% | 57.62% |
| **X-Guard** | **95.58%** | 99.28% | **92.88%** | **89.18%** | **91.53%** | **88.83%** | **43.70%** | 69.13% | **42.13%** | **69.69%** | **37.00%** | **65.55%** | **35.97%** | **63.51%** |
| **High-resource Languages (en, de, es, zh)** | | | | | | | | | | | | | | |
| *Token-level Watermark Method* | | | | | | | | | | | | | | |
| KGW | 97.05% | 99.89% | 71.75% | 75.05% | 59.13% | 62.43% | 0.15% | 55.85% | 0.15% | 52.17% | 0.25% | 54.27% | 0.05% | 41.90% |
| Uni-WM | 93.80% | 99.79% | 80.80% | 82.60% | 72.28% | 74.08% | 0.33% | 56.38% | 0.45% | 53.00% | 0.20% | 48.41% | 0.13% | 43.55% |
| SynthID | 99.45% | **99.90%** | 84.78% | 85.08% | 81.80% | 82.10% | 0.58% | 51.35% | 0.85% | 49.67% | 0.75% | 49.37% | 0.15% | 39.38% |
| *Sentence-level Watermark Method* | | | | | | | | | | | | | | |
| SemStamp | 86.50% | 96.90% | 84.48% | 86.88% | 82.30% | 83.10% | 33.18% | 66.35% | 31.10% | 57.40% | 35.43% | 63.78% | 33.00% | 56.80% |
| SimMark | 78.95% | 86.83% | 75.01% | 85.87% | 74.62% | 83.95% | 36.41% | 67.88% | 32.54% | 58.96% | 36.92% | 62.71% | 39.87% | 55.42% |
| **X-Guard** | **99.65%** | 99.80% | **96.43%** | **95.76%** | **93.85%** | **90.15%** | **49.61%** | **77.00%** | **49.18%** | **76.70%** | **49.49%** | **76.83%** | **47.15%** | **73.38%** |

approximated by the $p_{is}$ derived in Sec. 3.2. With a maximum of $T = 10$ attempts, SemStamp requires a theoretical minimum of **76 sentences** for reliable detection, whereas X-Guard requires only **7 sentences**. Although this gap narrows as $T$ increases (e.g., fewer than six sentences for both methods at $T = 100$), X-Guard achieves high detection reliability with fewer sentences and lower overhead.

# 4. Evaluation

## 4.1. Experimental Setup

**Tasks and Datasets.** We consider three generation settings: text continuation, question answering (QA), and text summarization. We select eight languages spanning: Arabic (ar), Chinese (zh), English (en), German (de), Spanish (es), Malay (ms), Hungarian (hu), and Thai (th). We randomly sample 500 prompts per language from CulturaX (Nguyen et al., 2024) and generate both watermarked and unwatermarked continuations, yielding 500 paired outputs per language. All baselines are evaluated on the raw model outputs without imposing any additional length constraints.

**Models Details and Attack Settings.** We choose LLaMA3.2-3B (Grattafiori et al., 2024), Gemma2-2B (Team, 2024), GPT-4o (OpenAI et al., 2024) as backbone, PySBD (Sadvilkar, 2020) as multilingual segmenter, as in Appendix A. We evaluate X-Guard under four attack strategies of increasing strength, details are in Appendix B.

**Metrics.** We use the Area Under the ROC Curve (AUC) and the Watermark Success Rate (WSR). AUC measures the verifier's ability to distinguish watermarked from unwatermarked texts across varying detection thresholds by sweeping the $z$-test threshold $Z_r$. WSR corresponds to the

true positive rate (TPR) of watermark detection under a fixed threshold, defined as WSR $= N_{detected}/N_{marked}$, where $N_{marked}$ is the number of watermarked texts and $N_{detected}$ denotes those whose $z$-scores exceed $Z_r$.

**Baselines.** Token-level baselines include KGW (Kirchenbauer et al., 2023), its robustness-enhanced variant Uni-WM (Zhao et al., 2024), and Google's sampling-based SynthID-Text (Dathathri et al., 2024). Sentence-level baselines include SemStamp (Hou et al., 2024a) and SimMark (Dabiriaghdam & Wang, 2025). For fair multilingual evaluation, we use the same base encoder for all methods.

**Implementation Details.** We use the multilingual embedding model BGE-m3 (Chen et al., 2024a) as the semantic encoder, whose hidden state dimension is 1024. The encoder is pretrained with a learning rate of $4 \times 10^{-5}$ and $\lambda = 3$, a maximum sentence length of 512, and a batch size of 4096, and is further fine-tuned for 5 epochs on a single NVIDIA H100 GPU. For generation, we use the first 50 tokens of each text as the prompt and apply A²PQ with a subspace dimension of $|\Omega| = 8$ ($M = 3$, $K = 2$) and a watermarked region ratio $\gamma = 0.25$. In accordance with Hou et al. (2024a), we set $T = 100$. The detection threshold for the $z$-test is set to $Z_r = 4$ throughout all experiments.

## 4.2. Evaluation on Reliability

**Overall Effectiveness.** From Table 1, X-Guard achieves consistently strong performance across both high- and low-resource languages. In the no-attack setting, it outperforms SemStamp by over 10 percentage points. Across all attack settings, X-Guard exceeds all baselines by 10–12 points on average. Under cross-lingual paraphrasing, it attains AUCs of 76.70% (high-res), compared to 57.40% for SemStamp,

*Table 2.* Performance across four downstream tasks on high- and low-resource languages using LLaMA3.2-3B.

| Model | ELI5 *Long-form QA* | | | FiQA *Finance QA* | | | MultiNews *Multi-Doc Summ* | | | QMsum *Query-Based Summ* | | |
|---|---|---|---|---|---|---|---|---|---|---|---|---|
| + Watermark | WSR | Rouge-L | DROP | WSR | Rouge-L | DROP | WSR | Rouge-L | DROP | WSR | Rouge-L | DROP |
| No Watermark (High-res) | - | 15.655 | - | - | 18.165 | - | - | 19.892 | - | - | 10.433 | - |
| + KGW | 0.995 | 14.732 | ↓5.9% | 0.990 | 17.744 | ↓2.3% | **0.850** | 18.782 | ↓5.6% | 0.395 | 9.368 | ↓10.2% |
| + Uni-WM | 0.995 | 14.194 | ↓9.3% | 0.955 | 15.940 | ↓12.3% | 0.670 | 18.582 | ↓6.6% | 0.435 | 9.066 | ↓13.1% |
| + SynthID-Text | 0.995 | 14.998 | ↓4.2% | 0.990 | 17.724 | ↓2.4% | 0.510 | 18.936 | ↓4.8% | 0.290 | 9.754 | ↓6.5% |
| + SemStamp | 0.842 | 15.233 | ↓2.7% | 0.891 | 17.823 | ↓1.9% | 0.582 | 19.422 | ↓2.4% | 0.337 | 9.917 | ↓4.9% |
| + SimMark | 0.943 | **15.456** | ↓1.3% | 0.855 | 17.234 | ↓5.1% | 0.657 | 19.468 | ↓2.1% | 0.352 | 10.035 | ↓3.8% |
| **+ X-Guard** | **0.998** | 15.407 | ↓1.6% | **0.993** | **18.286** | ↑0.7% | 0.830 | **19.786** | ↓0.5% | **0.530** | **10.281** | ↓1.5% |
| No Watermark (Low-res) | - | 8.669 | - | - | 10.737 | - | - | 11.586 | - | - | 6.686 | - |
| + KGW | **0.990** | 7.997 | ↓7.8% | 0.995 | 10.039 | ↓6.5% | 0.725 | 10.570 | ↓8.8% | 0.155 | 6.075 | ↓9.1% |
| + Uni-WM | 0.910 | 7.059 | ↓18.6% | 0.995 | 9.068 | ↓15.5% | 0.280 | 9.816 | ↓15.3% | 0.195 | 5.797 | ↓13.3% |
| + SynthID-Text | 0.920 | 7.907 | ↓8.8% | 0.995 | 10.645 | ↓0.9% | 0.090 | 11.009 | ↓5.0% | 0.070 | 6.056 | ↓9.4% |
| + SemStamp | 0.776 | **8.641** | ↓0.3% | 0.966 | 10.669 | ↓0.6% | 0.149 | 11.106 | ↓4.1% | 0.123 | 6.127 | ↓8.4% |
| + SimMark | 0.643 | 7.435 | ↓14.2% | 0.958 | 10.665 | ↓0.7% | 0.459 | 10.356 | ↓10.6% | 0.145 | 6.435 | ↓3.8% |
| **+ X-Guard** | **0.990** | 8.493 | ↓2.0% | **0.996** | **10.680** | ↓0.5% | **0.798** | **11.250** | ↓2.9% | **0.490** | **6.582** | ↓1.6% |

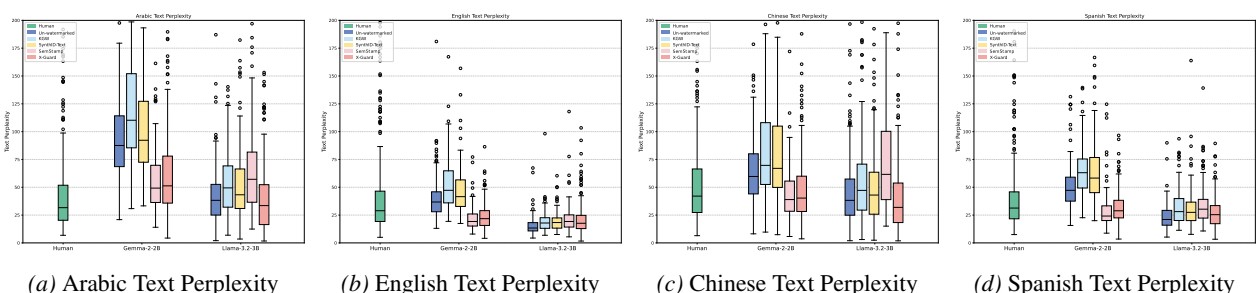

| *(a)* Arabic Text Perplexity | *(b)* English Text Perplexity | *(c)* Chinese Text Perplexity | *(d)* Spanish Text Perplexity |
|---|---|---|---|

*Figure 2.* A comparison of PPL across various watermarkings with different model sizes.

indicating that X-Guard better aligns watermark regions with the model's natural generation manifold.

**Robustness Against Adversarial Attacks.** We evaluate robustness under token-level substitution and sentence-level attacks in Table 1. Token substitution mildly affects sentence-level methods: under 30% substitution in low-res languages, SynthID degrades sharply (23.76 AUC), while X-Guard remains stable, improving over SemStamp by +9.26% AUC and +26.74% WSR. Sentence-level attacks introduce larger token shifts, causing token-level watermarks to nearly collapse (mean AUC 58.14%, WSR < 2%). X-Guard achieves the strongest robustness, attaining mean AUCs of 75.98% (high-res), surpassing SemStamp by +14.9%, indicating that it better preserves watermark signals on denser and more isotropic embedding manifolds.

### 4.3. Evaluation on Imperceptibility

**Perplexity.** We measure PPL on Gemma-2-2B and LLaMA-3.2-3B across four languages in Fig. 2. While Latin-based languages exhibit lower PPL than Arabic or Chinese, X-Guard consistently maintains low PPL across all languages and models, indicating minimal distortion of the model's native generation distribution.

**Downstream Task Performance.** All watermarkings incur

some degradation in Table 2, but X-Guard introduces the smallest mean drop: 0.7% / 1.75% on high-/low-res languages, substantially lower than baselines. QA tasks are generally more tolerant than summarization, as they focus on key answer tokens rather than sentence-level coverage. Despite increased sensitivity in summarization and low-res settings, X-Guard remains the most stable, achieving minimal utility loss (below 0.5% on QA) while retaining over 70% WSR in low-res languages.

### 4.4. Evaluation on Uniformity

**Cross-lingual Uniformity.** We evaluate languages spanning Latin (de, es, ms, en) and non-Latin scripts (ar, ru, th, zh). As shown in Table 3, non-Latin scripts exhibit slightly lower robustness: X-Guard maintains strong and stable performance across both groups (0.997 vs. 0.955) with substantially lower variance (std. 0.038 vs. 0.144 for SemStamp), indicating better alignment with heterogeneous writing systems. Unlike script differences, language resource level has minimal impact: LLaMA3.2-3B achieves nearly identical WSRs on high- and low-resource languages (0.978 vs. 0.974). Overall, X-Guard consistently outperforms SemStamp across all eight languages, with average gains of **+0.22** WSR and **+0.06** AUC.

*Table 3.* Cross-language comparison of SemStamp, SimMark, and X-Guard across different languages and multiple base models (LLaMA3.2-3B, Gemma2-2B, GPT-4o). Bold fonts mean the highest performance.

| Method | Arabic (ar) | | German (de) | | Spanish (es) | | Malay (ms) | | Chinese (zh) | | Thai (th) | | Hungarian (hu) | | English (en) | |
|---|---|---|---|---|---|---|---|---|---|---|---|---|---|---|---|---|
| | WSR | AUC | WSR | AUC | WSR | AUC | WSR | AUC | WSR | AUC | WSR | AUC | WSR | AUC | WSR | AUC |
| **LLaMA3.2-3B** | | | | | | | | | | | | | | | | |
| SemStamp | 0.785 | 0.929 | 0.860 | 0.963 | 0.821 | 0.962 | 0.798 | 0.959 | 0.846 | 0.966 | 0.485 | 0.835 | 0.524 | 0.860 | 0.933 | 0.985 |
| SimMark | 0.773 | 0.852 | 0.702 | 0.807 | 0.834 | 0.891 | 0.658 | 0.747 | 0.785 | 0.862 | 0.612 | 0.723 | 0.634 | 0.782 | 0.837 | 0.913 |
| **X-Guard** | **0.998** | **0.999** | **0.997** | **0.999** | **0.996** | **0.997** | **0.998** | **0.999** | **0.995** | **0.998** | **0.926** | **0.985** | **0.901** | **0.988** | **0.998** | **0.998** |
| **Gemma2-2B** | | | | | | | | | | | | | | | | |
| SemStamp | 0.934 | 0.981 | 0.944 | **0.986** | 0.882 | 0.972 | 0.900 | 0.979 | 0.904 | 0.983 | 0.507 | 0.847 | 0.513 | 0.848 | 0.945 | 0.990 |
| SimMark | 0.741 | 0.827 | 0.689 | 0.795 | 0.812 | 0.882 | 0.768 | 0.841 | 0.821 | 0.888 | 0.584 | 0.712 | 0.651 | 0.789 | 0.862 | 0.923 |
| **X-Guard** | **0.998** | **0.998** | **0.997** | **0.986** | **0.997** | **0.997** | **0.999** | **0.997** | **1.000** | **0.999** | **0.936** | **0.977** | **0.834** | **0.948** | **0.998** | **0.999** |
| **GPT-4o** | | | | | | | | | | | | | | | | |
| SemStamp | 0.770 | 0.934 | 0.843 | 0.945 | 0.818 | 0.969 | 0.782 | 0.920 | 0.812 | 0.923 | 0.390 | 0.814 | 0.621 | 0.869 | 0.904 | 0.968 |
| SimMark | 0.789 | 0.872 | 0.734 | 0.822 | 0.851 | 0.904 | 0.717 | 0.803 | 0.773 | 0.848 | 0.636 | 0.744 | 0.683 | 0.814 | 0.892 | 0.929 |
| **X-Guard** | **0.929** | **0.977** | **0.938** | **0.992** | **0.963** | **0.999** | **0.951** | **0.999** | **0.993** | **0.997** | **0.925** | **0.987** | **0.970** | **0.983** | **0.999** | **0.999** |

*Table 4.* Effect of different sentence encoders.

| Encoder | High-res | | Low-res | |
|---|---|---|---|---|
| | *No Attack* | *Attack Avg* | *No Attack* | *Attack Avg* |
| BGE-m3 + A&U Loss (X-Guard) | 0.997 | 0.643 | 0.956 | 0.572 |
| Frozen BGE-m3 | 0.996 | 0.627 | 0.955 | 0.568 |
| XLM-R (Conneau et al., 2020) | 0.995 | 0.616 | 0.953 | 0.552 |
| Qwen3-Embedding-0.6B (Zhang et al., 2025) | 0.996 | 0.641 | 0.955 | 0.566 |

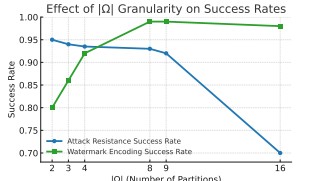

*(a)* Impact of $|\Omega|$ on WSR.

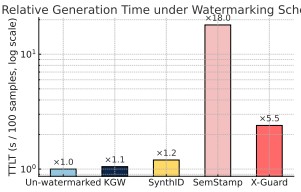

*(b)* Relative generation time.

*Figure 3.* Further analysis of X-Guard.

**Cross-model Uniformity.** We further evaluate X-Guard on open-source models and closed-source models. Across all models, X-Guard consistently outperforms SemStamp, improving WSR from 0.757→0.987 on LLaMA3.2 and 0.742→0.971 on GPT-4o. X-Guard remains highly stable, with a mean WSR standard deviation of only 0.041 compared to 0.155 for SemStamp.

### 4.5. Other Analysis

**Impact of Encoder Design.** We evaluate four encoders under both high- and low-resource settings (Table 4). Across all configurations, performance remains consistently high: non-attack WSR exceeds 0.955, and average attack-time performance stays above 0.64 (high-res) and 0.57 (low-res). Fine-tuning BGE-m3 with A&U loss yields consistent robustness gains over the frozen encoder, improving attack-time averages from 0.627→0.643 (high-res) without affecting non-attack performance.

**Watermarked Region Ratio.** $\gamma$ controls the fraction of $A^2PQ$ regions designated as watermarkable, affecting detection sensitivity and false positives. With $M = 3$ (yielding 8 regions), we evaluate $\gamma \in \{0.125, 0.25, 0.375, 0.5\}$, where $8\gamma$ is an integer. Experiments on 100 Chinese sentences from CulturaX reveal a clear trade-off: smaller $\gamma$ reduces false positives but weakens detectability, while larger $\gamma$ improves detection at the cost of increased false positives. Empirically, $\gamma = 0.25$ (2 of 8 regions) provides the best balance and is used as the default setting.

**Impact of Partition Granularity.** We vary the number of regions $|\Omega| \in \{2, 3, 4, 8, 9, 16\}$ (Fig. 3a). As $|\Omega|$ increases, the injection success rate improves sharply: from 0.80 at $|\Omega|=2$ to nearly 1.00 at $|\Omega|=8$ and then saturates, indicating diminishing returns from finer quantization. In contrast, robustness decreases with larger $|\Omega|$: coarse partitions (2–3 regions) achieve high robustness (above 0.95), while finer partitions increase boundary sensitivity, reducing robustness to around 0.70 at $|\Omega|=16$. Balancing these trade-offs, we select $|\Omega|=8$, which achieves near-perfect injection success while maintaining strong robustness (0.93).

**Efficiency of X-Guard.** We further analyze the theoretical overhead of $A^2PQ$, which incurs an $O(MK)$ per-step cost and an expected resampling bound of $1/\gamma$. To support our claim, we evaluate efficiency using relative generation time Time-To-Last-Token (TTLT) per 100 samples, normalized to the un-watermarked model (Fig. 3b). Sentence-level schemes suffer from rejection sampling, with SemStamp slowing down to ×18.0. X-Guard substantially improves sentence-level efficiency, reducing TTLT to ×5.5—over **3×** **faster** than SemStamp.

## 5. Conclusion

We propose a sentence-level watermarking framework **X-Guard** that achieves reliable, imperceptible, and universal detection. Through principled feature-space training

and partitioning, X-Guard demonstrates strong robustness against diverse real-world attacks while preserving semantic fidelity across models and languages.

## Impact Statement

This work aims to improve the reliability and robustness of sentence-level watermarking for AI-generated text, supporting content provenance and accountability. By enabling more consistent watermark detection across languages and deployment settings, it may help mitigate misuse such as misinformation and impersonation. We do not foresee significant negative societal impacts beyond those already well studied in prior work on LLM watermarking.

## Acknowledgments

This work is supported by the Postdoctoral Fellowship Program of CPSF under Grant Number GZC20251076, and the National Natural Science Foundation of China (No.U2336202).

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

# A. Model Settings

## A.1. Backbone Models

In this work, we utilize state-of-the-art (SOTA) models in the multilingual domain, including two open-source white-box models, Gemma2-2B and LLaMA3.2-3B, as well as the currently strongest closed-source black-box model, GPT-4o. Below, we provide a detailed description of each model.

- **Gemma2** model (Team, 2024) is the latest model proposed by Google in June 2024, incorporating several known technical modifications to the Transformer architecture, such as interleaving local-global attentions and group-query attention. To support multilingual tasks, Gemma2 employs a 256k vocabulary and supports over 100 languages. The model used in our experiments is a distilled version of the larger 7B model, and Gemma2-2B significantly outperforms other models of similar size.

- **LLaMA3.2-3B** model (Grattafiori et al., 2024) is part of the Meta Llama 3.2 collection of multilingual large language models (LMs) and was released in July 2024. This collection includes pretrained and instruction-tuned generative models in 1B and 3B sizes (text in/text out). The LLaMA3.2 instruction-tuned text-only models are optimized for multilingual dialogue use cases, including agentic retrieval and summarization tasks. These models outperform many open-source and closed-chat models on common industry benchmarks.

- **GPT-4o** model, released by OpenAI in late 2024, is one of the most advanced large language models in the industry. With state-of-the-art performance in both generation and understanding, GPT-4o excels in multilingual tasks and supports over 120 languages. Its robust multilingual capabilities allow it to handle a wide range of tasks, from language translation to complex reasoning in various languages. This model remains a black-box solution, not publicly available, but its exceptional performance across several benchmarks makes it a key choice in our experiments.

## A.2. Sentence tokenizer

We use the multilingual segmenter pySBD (Sadvilkar, 2020), whose language-specific rules provide stable cross-lingual behavior and robustness to noisy text. We also tested a punctuation-only tokenizer (commas/periods), which produces shorter segments and slightly improves AUC due to increased sentence counts. Nevertheless, we adopt sentence-level segmentation, as it better aligns with the natural generation of LLMs.

# B. Attack settings.

**Token-level substitution.** A lightweight surface attack that replaces tokens (10% and 30%) with synonyms or near-synonyms (keeping fluency) to disrupt lexical patterns exploited by token-level watermarks.

**Mono-lingual paraphrasing.** A stronger semantic-preserving attack that asks an external paraphraser to rewrite the text in the same language (e.g., "Paraphrase the following sentence in [language_name]") to remove stylistic cues while preserving meaning.

**Cross-language paraphrasing.** This attack leverages cross-lingual transformations to perturb sentence structure further: translate the text into a randomly chosen pivot language and back to the original language.

**Translation.** Translate across languages (e.g., English $\rightarrow$ Japanese translate) to exploit multilingual variation in tokenization and embedding projections.

**Optimization-based adaptive attack (strongest).** The adversary is assumed to know the watermarking algorithm and the encoder architecture, but not the secret key $k$. We re-implemented the DPO training in (Diaa et al., 2025) to train new paraphrasers (`Llama2-7b`) specifically against each watermarking method using corresponding positive/negative samples. Then use the paraphraser to find high-quality text variants that lie outside the watermarked regions.

For all cross-lingual and paraphrase-based attacks, we use `gpt-3.5-turbo-0613`[1] as the translator/paraphraser to ensure consistent and comparable transformations across languages and attack types.

---

[1] https://platform.openai.com/docs/models#gpt-3-5-turbo

## C. Downstream Tasks Datasets

Follow Waterbench (Tu et al., 2024), we utilize the following datasets:

- **Long-form QA** utilizes 200 samples from the ELI5 dataset (Fan et al., 2019), a long-form question-answering dataset originating from the Reddit forum "Explain Like I'm Five." **Rouge-L** is employed as the evaluation metric.

- **Finance QA** use 200 samples from the FinQA dataset (Maia et al., 2018). **Rouge-L** is employed as the evaluation metric.

- **Multi-Doc Summ** involves 200 samples from the widely-used MultiNews dataset (Fabbri et al., 2019), a multi-document news summarization dataset. **Rouge-L** serves as the evaluation metric.

- **Query-Based Summ** use 200 samples from the QMSum dataset (Zhong et al., 2021) with both input documents and queries for specific parts of the documents. **Rouge-L** is employed as the evaluation metric.

The original datasets are all in English, we use GPT-4o to translate them into the other seven languages together as Downstream Tasks Datasets.

