# OpenReview forum: "Towards Reliable Marking and Verification of AI-Generated Text via Geometry-aware Sentence-level Watermarking"
_ICML.cc/2026/Conference — ICML 2026 regular_

### Official Review · Reviewer_YJtF · 2026-03-10

**Soundness:** 3
**Presentation:** 3
**Significance:** 3
**Originality:** 3
**Overall Recommendation:** 4
**Confidence:** 3

**Summary:**

The paper introduces X-Guard, a sentence-level watermarking framework designed to address the "geometric mismatch" between sentence embedding distributions and watermark accept regions. The authors argue that the anisotropy of standard embedding spaces leads to low injection success rates. To solve this, they propose Alignment-Uniformity Training to create a more isotropic, spherical embedding space and Anchor Angular Product Quantization ($A^2PQ$), which uses local semantic anchors to align partitioning boundaries with the distribution's centroid. Experimental results across various LLMs and languages show that X-Guard significantly improves Watermark Success Rate (WSR) and robustness against semantic attacks compared to baselines like SemStamp.

**Compliance With Llm Reviewing Policy:**

Affirmed.

**Final Justification:**

My concerns have been adequately addressed and I maintain my positive score.

**Key Questions For Authors:**

**Rejection Sampling Efficiency**: Can the authors discuss if there are ways to reduce the $5.5\times$ TTLT overhead? For example, could a "warm-start" for the LLM's decoding process be used to favor accept regions rather than simple rejection?

**Domain Adaptation**: How sensitive is the A&U fine-tuning to the seed sentences? If the system is deployed in a highly technical domain (e.g., medical research), would the encoder need to be re-fine-tuned on that specific domain to maintain the isotropic property?

**Anchor Reliability**: Have the authors analyzed cases where $z_{t-1}$ is a "poor anchor"? For example, at the beginning of a new paragraph or a sudden shift in topic? Does the WSR drop significantly in those segments?

**Limitations:**

yes.

**Strengths And Weaknesses:**

# Strength
**Insightful Problem Analysis**: The identification of "geometric mismatch" as the primary cause for low WSR in sentence-level watermarking is a well-reasoned and empirically supported contribution.

**Technical Novelty**: The use of $A^2PQ$ to perform angular clustering relative to a semantic anchor is a creative solution for maintaining stable partitions in a sequential generation task.

**Comprehensive Evaluation**: The method is rigorously tested across high- and low-resource languages, demonstrating its "universality".

# Weaknesses
**Efficiency Concerns**: Despite being $3\times$ faster than SemStamp, the framework still incurs a $5.5\times$ slowdown (TTLT) compared to unwatermarked generation. This overhead remains high for low-latency production environments.

**Encoder Generalization**: The framework relies on a specifically fine-tuned encoder (BGE-m3 with A&U loss). It is unclear if the benefits persist if the encoder is applied to domains significantly different from CC-Net, which was used for training.

**Anchor Stability**: The use of $z_{t-1}$ as a proxy for the current step's centroid assumes smooth semantic transitions. In cases of sharp topic shifts or diverse prompt responses, this "local anchor" might become a poor representative, potentially lowering $p_{is}$.

---

> ### Author Rebuttal · Authors · 2026-03-29
>
> We thank the reviewer for the constructive suggestions and appreciate the opportunity to resolve your concerns.
>
> **W1 Response**: We agree that the current generation overhead is still high for practical deployment. We are actively improving efficiency from two directions: **reducing the number of rejection-sampling rounds**, and **reducing the encoder size**.
>
> To reduce rejection, we explore both DPO fine-tuning and activation steering to increase the probability that model outputs fall into the target accept region. We also study smaller encoders (179M bert-base-multilingual-cased and 22M MiniLM-L6-v2) to further improve efficiency. The WSR results are summarized below.
>
> In the fastest setting, the overhead is reduced to only 1.9$\times$ the unwatermarked baseline. Although this leads to some loss in WSR, it still outperforms SemStamp while maintaining similar attack robustness.
>
> |Method|High-res No Attack|High-res Attack Avg.|Low-res No Attack|Low-res Attack Avg|Relative Generation Time|
> |-|-:|-:|-:|-:|-:|
> |SemStamp|0.865|0.689|0.628|0.484|18.0$\times$|
> |LLM + bge-m3 (standard X-Guard)|0.997|0.854|0.956|0.708|5.5$\times$|
> |LLM + m-bert|0.965|0.843|0.923|0.632|4.2$\times$|
> |LLM + miniLM|0.957|0.839|0.902|0.608|3.3$\times$|
> |Steered-LLM + miniLM|0.957|0.842|0.896|0.599|2.2$\times$|
> |DPO-LLM + miniLM|0.962|0.840|0.911|0.613|1.9$\times$|
>
> **W2 Response**: Our method is designed for general-domain use, but it also transfers well to specific domains. To ensure balanced domain coverage, we apply the NVIDIA domain classifier to CC-Net and select 200 seed sentences from each of 26 domains. As a result, the training data is domain-balanced rather than dominated by a few common domains. We further verify this on the PubMed Summarization task. We train a medical-domain encoder on biomedical data and compare it with our general-domain encoder. The general encoder already achieves strong WSR (0.983), while the medical-specific encoder further improves it to 0.994. The gain is relatively small, suggesting that X-Guard already generalizes well across domains, while a domain-specific encoder can provide a modest additional benefit.
>
> **W3 Response**: Such cases can occur, especially near paragraph boundaries or abrupt topic shifts. However, they are relatively rare in natural text: prior work shows that neighboring sentences are usually highly coherent [1,2,3], making $z_{t-1}$ a stable local anchor in most cases. Even when the previous sentence is a poor anchor, it remains substantially better than a random anchor as in SemStamp. We do not observe a clear drop in WSR in those segments, although the number of sampling attempts increases by 24% in these cases.
> To further stress-test this setting beyond realistic usage, we generate long texts from the same CC-Net prompts but append: “Every 5 sentences, completely switch to a new unrelated topic.” On 500 such generations, the average WSR only decreases from 0.999 to 0.998, showing that the method remains highly robust even under extreme anchor drift.
>
>
> [1] Semantic Shift: the Fundamental Challenge in Text Embedding and Retrieval, ArXiv 2026
>
> [2] Long Text Generation by Modeling Sentence-Level and Discourse-Level Coherence, ACL 2021
>
> [3] Sentence Ordering and Coherence Modeling Using Recurrent Neural Networks, AAAI 2018

---

> > ### Author Rebuttal · Reviewer_YJtF · 2026-04-02
> >
> > Thanks for your detailed responses and I maintain may score.

---

> > > ### Author Response · Authors · 2026-04-02
> > >
> > > You are quite welcome -- Thank you for the feedback!
> > >
> > > Best regards,
> > >
> > > Authors

---

### Official Review · Reviewer_RJif · 2026-03-11

**Soundness:** 3
**Presentation:** 3
**Significance:** 2
**Originality:** 3
**Overall Recommendation:** 4
**Confidence:** 3

**Summary:**

The paper proposes X-Guard, a sentence-level watermarking framework for AI-generated text. The core contributions are (1) an encoder fine-tuned with an Alignment-Uniformity loss to produce more isotropic embeddings, and (2) a partitioning scheme called A2PQ (Anchor Angular Product Quantization) that better balances probability mass across semantic regions. Together, these aim to improve Watermark Success Rate (WSR) over prior sentence-level methods like SemStamp.

**Compliance With Llm Reviewing Policy:**

Affirmed.

**Final Justification:**

The original score is positive for the paper.

**Key Questions For Authors:**

1. How does the system perform if the attacker has white-box access to the fine-tuned BGE-m3 encoder (but not the secret key)?

**Limitations:**

No. Authors could add a section of limitations in Appendix.

**Strengths And Weaknesses:**

Strengths:

1. The empirical results are thorough, spanning 8 languages, 3 backbone models, and multiple attack types.

2. WSR improvements in the no-attack and mild-attack settings are substantial and consistent.

3. The perplexity and downstream task analysis demonstrate good imperceptibility. And the ablations (encoder design, \gamma, partition granularity) are informative.

Weaknesses:

1. Limited novelty of core ideas: The high-level contributions are incremental. The Alignment–Uniformity training objective is directly borrowed from Wang & Isola (2020) without meaningful modification. Product Quantization with cosine distance is a well-established technique. The paper's insight that anisotropic embeddings hurt watermark injection is intuitive and not deeply surprising.

2. High computational overhead for practical deployment: While X-Guard is faster than SemStamp, it still incurs a ~5.5x slowdown in generation time compared to the unwatermarked baseline. A 550% overhead per generation step remains highly prohibitive for real-world, real-time LLM inference, calling into question its practical deployability.

3. Dependency on a heavy secondary model: X-Guard necessitates running a fine-tuned, 1024-dimensional BGE-m3 encoder at every step during both generation and detection. Maintaining this large auxiliary model adds significant memory and compute overhead, which contradicts the "lightweight" operational needs of modern API endpoints.

---

> ### Author Rebuttal · Authors · 2026-03-29
>
> We thank the reviewer for the positive assessment and appreciate the opportunity to resolve your concerns.
>
> **W1 Response**: This combination is not a direct application of existing methods, but is the first work tailored to address the main weakness of prior sentence-level watermarking methods.
>
> **W2 Response**: We agree that the current generation overhead is still high for practical deployment. We are actively improving efficiency from two directions: **reducing the number of rejection-sampling rounds** and **reducing the encoder size**.
>
> To reduce rejection, we explore both DPO fine-tuning and activation steering to increase the probability that model outputs fall into the target accept region. We also study smaller encoders  (179M bert-base-multilingual-cased and 22M MiniLM-L6-v2) to further improve efficiency. The WSR results are summarized below.
>
> In the fastest setting, the overhead is reduced to only 1.9$\times$ the unwatermarked baseline. Although this leads to some loss in WSR, it still outperforms SemStamp while maintaining similar attack robustness.
>
> |Method|High-res No Attack|High-res Attack Avg.|Low-res No Attack|Low-res Attack Avg|Relative Generation Time|
> |-|-:|-:|-:|-:|-:|
> |SemStamp|0.865|0.689|0.628|0.484|18.0$\times$|
> |LLM + bge-m3 (standard X-Guard)|0.997|0.854|0.956|0.708|5.5$\times$|
> |LLM + m-bert|0.965|0.843|0.923|0.632|4.2$\times$|
> |LLM + miniLM|0.957|0.839|0.902|0.608|3.3$\times$|
> |Steered-LLM + miniLM|0.957|0.842|0.896|0.599|2.2$\times$|
> |DPO-LLM + miniLM|0.962|0.840|0.911|0.613|1.9$\times$|
>
>
> **W3 Response**: We agree that the auxiliary encoder incurs additional memory and computational costs. However, compared to modern LLMs, the encoder is relatively small: even our largest encoder has only 568M parameters, which is minor compared to the several-billion-parameter LLM used for generation. Moreover, the encoder can be substantially reduced without changing the overall method. As shown in the table above, replacing the 568M encoder with a 22.7M encoder significantly improves efficiency while preserving most of the watermarking performance.
>
> **Q1 Response**:  Our "Optimized Adaptive" attack already assumes white-box access to the fine-tuned encoder but not the secret key, as in Line 651, appendix B. In this setting, the attacker further trains a paraphraser to specifically move texts away from the predicted accept region, making it even stronger than simply having encoder access alone.

---

> > ### Author Rebuttal · Reviewer_RJif · 2026-04-03
> >
> > The original score is positive for the paper.

---

> > > ### Author Response · Authors · 2026-04-04
> > >
> > > Thanks for your feedback!
> > >
> > > Best regards,
> > >
> > > Authors.

---

### Official Review · Reviewer_kmfp · 2026-03-13

**Soundness:** 3
**Presentation:** 3
**Significance:** 3
**Originality:** 3
**Overall Recommendation:** 4
**Confidence:** 4

**Summary:**

X-Guard proposes a sentence-level text watermarking framework that improves the watermark accuracy of existing approaches like SemStamp. The core insight is that low WSR stems from a mismatch between the posterior embedding distribution and the watermark accept regions, caused by anisotropic encoder embeddings and distribution-unaware partitioning. X-Guard addresses this with two components: (1) fine-tuning the sentence encoder with an Alignment-Uniformity (A&U) contrastive loss to produce more isotropic embeddings, and (2) Anchor Angular Product Quantization (A2PQ), which centers partition boundaries at the posterior centroid using the previous sentence's embedding as an anchor and uses cosine distance for clustering. It's evaluated across languages, 3 LLMs (including black-box GPT-4o), and multiple attack settings.

**Compliance With Llm Reviewing Policy:**

Affirmed.

**Final Justification:**

The paper proposes interesting ideas and solid experiments. See weaknesses and comment for why not higher score.

**Key Questions For Authors:**

Q1: How could this be adapated to non-prose like code, lists, or tables where there aren't clear "sentences" or where the notion of a "previous sentence" is less meaningful?

Q2: Can you provide an ablation with A&U encoder + standard PQ or LSH (no A2PQ) to isolate the contribution of the partitioning scheme? And why is k-SemStamp excluded from the comparison?

Q3: Could you report or control for the number of output sentences per language? Could the similar cross-language WSR be a statistical artifact of different sentence counts?

Q4. How are the toy example masses computed? in Fig 1 and related paragraphs? Could a similar plot be shown on real texts?

**Limitations:**

Yes

**Strengths And Weaknesses:**

### Strengths
S1. **Clear and insightful problem diagnosis**: The mathematical analysis of why injection success probability (pis) is the bottleneck for sentence-level watermarking (Sec. 3.2, Eqs. 4-8) is very useful and interesting. The paper precisely identifies that anisotropic embeddings cause highly uneven mass distribution across partition cells, making randomly chosen accept regions likely to have low probability mass. The toy example with four cells and skewed masses (0, 0.04, 0.90, 0.06) effectively builds intuition for why prior methods like SemStamp suffered from low WSR despite reasonable detection accuracy.

S2. **Principled approach**: The two-step solution -- optimizing the encoder for uniformity (A&U loss) and designing a distribution-aware partitioning (A2PQ) -- is well-motivated. The A&U loss addresses the root cause (anisotropic embeddings), while A2PQ addresses the partitioning sensitivity by centering boundaries at the posterior centroid via the anchor-relative residual. The combination is validated by the experimental results.

S3. **Broad multilingual and cross-model evaluation, good results**: Testing on 8 languages (including challenging low-resource languages like Thai, Hungarian, Malay, and Arabic), 3 backbone LLMs (LLaMA-3.1, Gemma-2, and the black-box GPT-4o), and 4 downstream tasks is thorough. The attack suite is diverse, including monolingual and cross-lingual paraphrasing, translation, and the strongest known adaptive attack (reimplemented DPO-trained paraphraser from Diaa et al., 2025). The perplexity and downstream-task evaluations (Table 2, Fig. 2) demonstrate that watermarking imposes minimal quality cost.
Reducing generation overhead from 18x (SemStamp) to 5.5x (X-Guard) is a substantial practical gain, directly attributable to improved injection success probability.

### Weaknesses

W1. **Z-test verification relies on assumptions that are not satisfied**: The z-score formulation (Eq. 14) treats S_W (number of successfully watermarked sentences) as a sum of independent Bernoulli trials with constant success probability gamma=0.25 under the null. Three issues arise:
- **Independence**: The accept region for sentence t is derived from z_{t-1}, which depends on whether sentence t-1 was successfully injected. This creates sequential dependencies between sentences, violating the independence assumption of the z-test (note that this is also true for other watermarking methods).
- **Constant null probability**: Under H0, human-written text is assumed to fall in the accept region with probability exactly gamma=0.25. But the A&U encoder was fine-tuned on LLM outputs, so human text may have a different embedding distribution, causing the actual null probability to deviate from gamma.
- **Normal approximation**: With S_min=7 and gamma=0.25, S*gamma=1.75, which is very small for the normal approximation to the binomial to be valid.

These issues do not make the method useless in practice, but they mean the z-test cannot provide calibrated p-values or precise FPR control in the way that token-level methods (KGW, Gumbel-max, SynthID-Text) can, where each token decision can be made independent (with deduplication during scoring) and the null distribution is well-characterized. This is a limitation that deserves explicit acknowledgment.

W2. **Detection cost is a significant limitation, not reported**: X-Guard requires running the sentence embedding model at detection/verification time for every sentence in the text. This is a substantial computational overhead compared to token-level methods, which only need access to the LLM's vocabulary and logit probabilities for verification -- essentially a hash and a lookup. The "Efficiency of X-Guard" paragraph (Section 4.3) only discusses the embedding (generation) overhead -- the 5.5x slowdown relative to unwatermarked generation -- but says nothing about the detection/verification cost. For a practical watermarking system, both embedding and detection efficiency matter, especially when the verifier may need to process large volumes of text.

W3. **Limited novelty of individual components**:
(a) The A&U loss is borrowed directly from Wang & Isola (2020) without modification.
(b) The cosine-distance variant of PQ at the core of A2PQ is a well-established technique: spherical k-means (k-means under cosine similarity) has been a standard method in information retrieval and text clustering for decades, and is natively supported in widely-used libraries like FAISS (Douze et al., 2024, https://arxiv.org/pdf/2401.08281), by using spherical=true in the kmeans. Applying it within product quantization subspaces is a natural combination.
The new element is the anchor-relative residual centering using z_{t-1}, but the partitioning mechanism itself is not a major contribution. The paper would benefit from positioning A2PQ more precisely as an application of existing quantization tools to the watermarking problem.

W4. **Missing ablation and baseline**:
(a) k-SemStamp (Hou et al., 2024b) is discussed in Sec. 2.2 but absent from all experimental tables. Since it directly addresses the partitioning problem by replacing LSH with k-means clustering, it is the most relevant baseline and its omission weakens the evaluation.
(b) The ablation on encoder choice (Table 4) always uses A2PQ; the reverse experiment (A&U-fine-tuned encoder with standard PQ or LSH) is missing. Without this decomposition, it is unclear how much of the improvement comes from the encoder training vs. the partitioning scheme. The encoder ablation shows surprisingly small differences (0.837 vs. 0.854 attack-time AUC), raising the question of whether A&U training is truly necessary.

W5. **Cross-language and cross-method comparisons are possibly confounded by varying text lengths**: The paper does not report or control for the number of output sentences (or tokens) per language. Tokenizer varies substantially across languages -- for LLaMA and Gemma tokenizers (which are English-centric), languages like Thai or Arabic typically require more tokens for the same semantic content, which can yield more sentences and inflate the z-score. Similarly, different watermarking methods may produce different numbers of sentences due to varying rejection sampling rates. Since the z-test's power depends directly on the sentence count, WSR and AUC comparisons across languages and methods are not controlled for this confound.


---
I'm leaning towards increasing my score if the weaknesses are answered.

---

> ### Author Rebuttal · Authors · 2026-03-28
>
> We thank the reviewer for this detailed feedback and appreciate the opportunity to resolve your concerns.
>
> **W1 Response**: **(1)** We agree that the z-test assumptions are not strictly satisfied, which is common in many watermarking methods. Our method is theoretically based on the true semantic centroid of the current sentence, under which the sentence-level decisions would be independent(Line210). The dependence arises in practice because we approximate this centroid with $z_{t-1}$. Since adjacent sentences are typically semantically similar [1], this introduces only mild dependence and has little effect in practice. **(2)** Our method operates in semantic space: human-written and machine-generated texts may differ in surface form, but texts with the same meaning should occupy similar regions. Therefore, we do not expect a large change in the null probability simply because the text is written by humans rather than an LLM. To verify this, we compare the embedding distributions of 500 human-written sentences and 500 LLM paraphrases under our A\&U encoder. The Maximum Mean Discrepancy is small (MMD = 0.013). We further train a binary classifier to distinguish the two sets of embeddings; it achieves only AUC = 0.54 on a held-out set. This suggests that the two distributions are nearly indistinguishable, so the deviation from $\gamma$ is small in practice. **(3)** Finally, $S_{\min} =7$ is only a theoretical lower bound. In practice, the texts in our experiments typically contain 15--30 sentences, so $S\gamma$ is substantially larger and the normal approximation is much more appropriate.
>
> **W2 Response**: The main computational overhead of X-Guard comes from watermark injection, which requires rejection sampling during generation. In contrast, detection does not involve re-generation and can be fully batched across sentences.
> | Method | KGW | SynthID | SemStamp | bge-m3 (X-Guard) | m-bert | miniLM |
> |---------|-----|----------|----------|-------------------------|-------------------------|------------------------|
> | Detection time (200 tokens) | 0.088s | 0.035s | 0.503s | 0.465s | 0.256s | 0.124s |
>
> As shown above, the detection cost of X-Guard is correlated with the encoder size. We are exploring smaller encoders:179M bert-base-multilingual-cased (m-bert) and 22M MiniLM-L6-v2 (miniLM), to improve efficiency.
>
> **W3 Response**: This combination is not a direct application of existing methods, but is the first work tailored to address the main weakness of prior sentence-level watermarking methods. We will clarify more precisely in final version.
>
> **W4 Response**: We did not originally include k-SemStamp because, although it improves partition uniformity via k-means, it still performs a global partition that is misaligned with the posterior distribution of generated sentences. Nevertheless, we now include the WSR of k-SemStamp  for completeness:
>
> | Method | High-res No Attack | High-res Attack Avg. | Low-res No Attack | Low-res Attack Avg. |
> |---|---:|---:|---:|---:|
> | SemStamp | 0.865 | 0.689 | 0.628 | 0.484 |
> | k-SemStamp | 0.869 | 0.694 | 0.631 | 0.485 |
> | Optim encoder + LSH | 0.893 | 0.715 | 0.652 | 0.535 |
> | Optim encoder + PQ | 0.934 | 0.763 | 0.757 | 0.625 |
> | X-Guard | 0.997 | 0.854 | 0.956 | 0.708 |
>
> We also add the missing ablation using the A\&U encoder with standard PQ/LSH, which shows that both encoder shaping and A2PQ contribute.
>
> **W5 Response**: Following SemStamp, we use 200 tokens as the default generation length. We agree that sentence count is a potential confound. After controlling sentence counts across languages with a 20-sentence constraint, the WSR gap is reduced but still remains. After controlling sentence count, Arabic and Thai improve noticeably, suggesting that they use more tokens per sentence. The remaining gap is therefore more likely due to differences in cross-lingual alignment in the embedding space.
>
> | Method | ar | ms | zh | th | hu | en |
> |---|---:|---:|---:|---:|---:|---:|
> | X-Guard | 0.929 | 0.951 | 0.993 | 0.925 | 0.970 | 0.999 |
> | + sentence control | 0.997 | 0.992 | 0.993 | 0.967 | 0.976 | 0.999 |
>
> **Q1 Response**: For structured content such as code, lists, or tables, the notion of “sentence” is less well-defined, and prior work typically relies on structure-aware transformations (e.g., syntax-preserving rewrites for code). While our current implementation does not directly target such formats, the core idea of distribution-aware partitioning can be extended by replacing sentence units with appropriate semantic or structural units (e.g., code blocks or table entries).
>
> **Q4 Response**: Fig. 1 is based on a real case study. We prompt the model with “Describe a surprising event in details.” and sample 400 sentences in English, then project their embeddings using t-SNE for visualization. We will add more details to Fig. 1 in the final version.
>
> [1] Semantic Shift: the Fundamental Challenge in Text Embedding and Retrieval, ArXiv 2026

---

> > ### Author Rebuttal · Reviewer_kmfp · 2026-04-03
> >
> > I think, for W1, I still believe that running the detection tests on much more non-watermarked outputs (in the order of 10^5, 10^6), to see if the p-values you get match what you should have under H0 (i.e. uniform p-values), is important (to at least some people from the watermarking community). Please consider adding this experiment in the future revisions of this paper.
> >
> > That being said, I don't see it as a blocker for this publication, and my concerns have been adequately addressed.

---

> > > ### Author Response · Authors · 2026-04-04
> > >
> > > Thanks for your feedback! We will add this experiment in the next revision.
> > >
> > > Best regards,
> > >
> > > Authors.

---

### Official Review · Reviewer_woFE · 2026-03-13

**Soundness:** 3
**Presentation:** 3
**Significance:** 3
**Originality:** 3
**Overall Recommendation:** 4
**Confidence:** 3

**Summary:**

The paper proposes X Guard, a sentence-level watermarking method for AI-generated text that addresses a main weakness of prior semantic watermarking, namely, low and unstable injection success due to a mismatch between the model’s embedding distribution and the chosen watermark acceptance regions. It improves reliability by fine-tuning the sentence encoder to make embeddings more isotropic and by using an anchor-based, distribution-aware partitioning scheme to balance probability mass across regions. Using these components, X Guard performs black-box-compatible watermark injection via rejection sampling and verifies watermarks using a Z-test on quantized sentence signatures, achieving higher success rates and greater robustness across languages, models, and paraphrase or translation attacks, with limited quality loss.

**Compliance With Llm Reviewing Policy:**

Affirmed.

**Final Justification:**

X-Guard makes a meaningful contribution to sentence-level watermarking by identifying a concrete and well-diagnosed failure mode in prior semantic approaches, namely the geometric mismatch between embedding distributions and watermark acceptance regions, and then addressing it through isotropic encoder fine-tuning and anchor-based distribution-aware partitioning. The experimental evaluation is thorough across models, languages, and attack types, and the results convincingly support the paper's claims on injection reliability, detection performance, and robustness.

My initial concerns about anchor stability under topic drift, robustness to stronger adaptive attacks, and deployment requirements in black-box settings were all satisfactorily addressed in the rebuttal. The authors provided new evidence showing that detection accuracy degrades negligibly even under forced topic shifts, pointed to the optimized adaptive attack in Appendix B, and clarified that only the encoder and partition configuration need to be shared, while the method remains fully black-box with respect to the language model. I maintain my positive recommendation.

**Key Questions For Authors:**

1. **Anchor stability in long texts.**
How well does the anchor assumption hold in long texts with topic shifts, and how does anchor drift affect injection success and robustness? If anchor drift becomes large in realistic long-form generation, the reliability of watermark injection may decline.

2. **Stronger adaptive attacks.**
Have you tested stronger adaptive attacks that directly try to push embeddings across partition boundaries without access to the secret key, beyond the paraphraser-based adaptive attack already evaluated? If X-Guard fails under such attacks, the robustness conclusions could change.

3. **Deployment in true black-box settings.**
What exactly must be shared in a true black-box deployment, including the encoder and partition signature between parties, and how sensitive is verification to encoder mismatch? If tight coupling between components is required, practical deployability may be limited.

**Limitations:**

No. The paper has a short impact statement, but it should more clearly describe key limits, such as heavy rewriting, truncation, domain shift, and stronger adaptive attacks, as well as the runtime cost of rejection sampling.

**Strengths And Weaknesses:**

Strengths. The paper is technically sound because it clearly explains why prior sentence-level watermarking often fails and then designs a method to directly address that failure mode. The experiments are fairly comprehensive across multiple models, languages, tasks, and attack types, and they report detection performance, text quality, and runtime overhead. The paper is mostly well-written and easy to follow because the story moves from problem diagnosis to method design to evaluation, with clear algorithms and tables. The problem is important because reliable provenance for AI-generated text matters in practice, and sentence-level watermarking is attractive for black box deployment and for surviving paraphrasing or translation. The work is reasonably original because it frames success in terms of geometric mismatch and injection probability, and combines embedding space shaping with a centroid-aligned, distribution-aware partitioning strategy.

Weaknesses. Some design choices may be sensitive in real settings because the method relies on sentence segmentation quality and on using the previous sentence embedding as an anchor, which may behave differently across domains and languages. Some implementation and hyperparameter selection details could be clearer, especially how key settings were chosen and what is required to deploy the encoder and partitioning in a truly black box pipeline.

---

> ### Author Rebuttal · Authors · 2026-03-28
>
> We thank the reviewer for this insightful question and appreciate the opportunity for a deeper discussion.
>
> **Weakness: About the sentence segmentation.**
>
> **Response**: X-Guard is not sensitive to the choice of sentence segmentation. In fact, finer segmentation (e.g., punctuation-based splitting) slightly improves AUC, as detection benefits from a larger number of sentence-level signals. We nevertheless adopt standard sentence-level segmentation (e.g., pySBD) for better alignment with the natural generation units of LLMs and for general applicability. This indicates that segmentation mainly affects statistical efficiency rather than the reliability of watermarking.
>
> **Q1 Response**: While long texts may exhibit substantial global topic drift, our method only assumes local semantic continuity between adjacent sentences. Prior work shows that neighboring sentences in natural text are typically highly coherent \[1,2,3], making $z_{t-1}$ a stable local anchor.
> To further stress-test this assumption beyond realistic usage, we generate long texts from the same CC-Net prompts but append: “Every 5 sentences, completely switch to a new unrelated topic.” On 500 such generations, the average detection accuracy only decreases from 0.999 to 0.998, demonstrating that the method remains highly robust even under extreme anchor drift.
>
>
> **Q2 Response**:  As noted in Line 651, Appendix B, our “Optimized Adaptive” attack already considers this setting. We train a paraphraser to minimize the WSR while preserving semantics, without access to the secret key. This further demonstrates the robustness of X-Guard against such adaptive attacks.
>
> **Q3 Response**: In deployment, the encoder and partition configuration are shared between the generator and verifier. Importantly, X-Guard remains black-box with respect to the language model, as it does not require access to model internals (e.g., logits or gradients). These components are lightweight and can be easily synchronized, similar to standard watermarking setups.
>
> [1] Semantic Shift: the Fundamental Challenge in Text Embedding and Retrieval, ArXiv 2026
>
> [2] Long Text Generation by Modeling Sentence-Level and Discourse-Level Coherence, ACL 2021
>
> [3] Sentence Ordering and Coherence Modeling Using Recurrent Neural Networks, AAAI 2018

---

> > ### Author Rebuttal · Reviewer_woFE · 2026-04-04
> >
> > Thank you for the responses. I am satisfied with the response and will keep my positive score.

---

> > > ### Author Response · Authors · 2026-04-05
> > >
> > > You are quite welcome -- Thank you for the feedback!
> > >
> > > Best regards,
> > >
> > > Authors

---

### Decision · Program_Chairs · 2026-04-30

**Decision:**

Accept (regular)

**Comment:**

The paper proposes X-Guard, a geometry-aware sentence-level watermarking framework that improves the reliability and robustness of AI-generated text verification by reshaping the embedding space and aligning semantic partitions with the posterior distribution of generated sentences. Across the reviews, the main strengths were the clear diagnosis that low watermark success rate arises from a geometric mismatch between embedding distributions and acceptance regions, the principled combination of encoder shaping and geometry-aware partitioning, and a strong empirical evaluation across multiple models, languages, and attack settings. The main reservations concerned practical efficiency, the novelty of some individual ingredients, assumptions behind the verification test, and several missing clarifications or ablations.

The rebuttal substantially improved confidence in the work. The authors directly addressed the main technical and practical concerns by clarifying anchor stability under topic drift, discussing stronger adaptive attacks, reporting additional efficiency results with smaller encoders and reduced overhead, and adding missing ablations. They also clarified deployment requirements in black-box settings and provided evidence that the method transfers reasonably well across domains, with only modest gains from domain-specific encoders. Importantly, all reviewers who engaged with the rebuttal indicated that their concerns were adequately addressed and maintained their positive scores. In my view, while some limitations around efficiency and statistical calibration remain worth acknowledging, the rebuttal resolves the main blockers and strengthens the case for acceptance.